# Recent Advances in Porphyrin-Based Inorganic Nanoparticles for Cancer Treatment

**DOI:** 10.3390/ijms21093358

**Published:** 2020-05-09

**Authors:** Hanieh Montaseri, Cherie Ann Kruger, Heidi Abrahamse

**Affiliations:** Laser Research Centre, Faculty of Health Sciences, University of Johannesburg, P.O. Box 17011, Doornfontein 2028, South Africa; montaseri.hanieh@gmail.com (H.M.); cherier@uj.ac.za (C.A.K.)

**Keywords:** Porphyrins, nanotheranostics, inorganic nanoparticles, cancer treatment

## Abstract

The application of porphyrins and their derivatives have been investigated extensively over the past years for phototherapy cancer treatment. Phototherapeutic Porphyrins have the ability to generate high levels of reactive oxygen with a low dark toxicity and these properties have made them robust photosensitizing agents. In recent years, Porphyrins have been combined with various nanomaterials in order to improve their bio-distribution. These combinations allow for nanoparticles to enhance photodynamic therapy (PDT) cancer treatment and adding additional nanotheranostics (photothermal therapy—PTT) as well as enhance photodiagnosis (PDD) to the reaction. This review examines various porphyrin-based inorganic nanoparticles developed for phototherapy nanotheranostic cancer treatment over the last three years (2017 to 2020). Furthermore, current challenges in the development and future perspectives of porphyrin-based nanomedicines for cancer treatment are also highlighted.

## 1. Introduction

Despite some major breakthroughs in cancer treatment, cancer is still one of the most significant causes of death in patients, after heart and infectious diseases [1]. Current disadvantages of conventional cancer therapies, such as Chemotherapy and Radiation, are nonspecific drug delivery mode of actions, poor bio-distribution, and biological barriers clearance, which limit their overall effectiveness [2,3]. Nanotechnology can overcome these obstacles by using engineered nanomedicines, such as nanoparticles (NPs) conjugated drugs. Further to this, cancer nanotheranostics has emerged as a promising research area, where nanotechnology is used to integrate the diagnosis and treatment of cancer by combining nanoplatforms with therapeutic agents, in order to enhance tumor specific targeted drug accumulation in cancer cells only, leaving the normal cells unaffected [4].

Various NPs have been fabricated for cancer diagnostics and therapeutics which have the ability to allow the drug to by-pass the immune system, as well as allow drugs to passively accumulate in cancerous tissues [1]. It has been demonstrated that the relationship between NPs and innate immune system responses (such as antigen-presenting cells or macrophage) depends on various parameters including size, shape, surface modification and charge, as well as hydrophobicity of NPs [5,6]. For example, PEGylation of NPs can prevent non-specific uptake by innate immune system [7]. More importantly, therapeutic NPs require having immunosuppressive or anti-inflammatory properties to phagocytes in order to decrease the therapeutic dose and immune-based side effects [5,8]. For this reason, NPs conjugated with targeting ligands, such as antibodies show prolonged immunosuppressive effects [9] and improved specificity to cancer cells which consequently declines off-target accumulation of drugs [1].

A variety of biomedical therapies including, but not limited to photodynamic therapy (PDT) and photothermal therapy (PTT) have been developed to surmount the series of obstacles experienced within cancer treatment research. PDT and PTT are non-invasive types of phototherapies, which exhibit lower side effects, when compared to conventional cancer therapies which have very high toxicity in normal tissues [10]. Photosensitizers (PSs), specific wavelength of light and molecular oxygen are indispensable parts of phototherapy. PSs absorb light energy to generate reactive oxygen species (ROS) for PDT [11] or use photon energy released as molecular vibrations to produce PTT hyperthermia [12].

However, the depth of treatment relies on the wavelength of light that can activate a PS to generate reactive oxygen species (ROS). Shorter wavelengths (<650 nm) have a low penetration depth in tissues, while longer wavelengths above 850 nm are not sufficient for PS activation to generate excited singlet oxygen [13]. Therefore, the most appropriate phototherapeutic window index wavelength of PDT is within 650–850 nm range [13]. In addition, near infrared (NIR) light is the most suitable region in PTT treatment. In particular, NIR-II window (1000–1350 nm), compared to NIR-I window (750–950 nm) [14], since it can provide deep-seated tumor tissue penetration with high maximum permissible exposure to lasers (i.e., 1 W/cm^2^ at 1064 nm laser, 0.33 W/cm^2^ at 808 nm laser) [15].

Amongst the different generations of PS available today, current research seems to focus on the development or investigation into various porphyrin PS types. Porphyrin PSs and their derivatives are organic heterocyclic macrocycles with a high phototoxicity [16] and can be applied in multiple diagnostic and therapeutic functions. However, porphyrins exhibit low water solubility and self-aggregation properties, which cause major challenges in PS subcellular localization and uptake, thus affecting the overall treatment outcomes in PDT and PTT cancer therapy [17]. Furthermore, these phototherapies allow porphyrins to combine with other therapeutic modalities in a highly adjuvant-dependent manner [18]. Therefore, many researches have shifted their focus into porphyrin-based nanomedicines [18,19], since nanomedicines can enhance the bioavailability of therapeutic agents and so assist them to accumulate passively in tumors via the enhanced permeability and retention (EPR) effect [20]. Lastly, the multifunctionality of nanomedicines is of great value in order to assist in alleviating unwanted cancer treatment side effects [1].

## 2. Conventional and Unconventional Cancer Treatments

Cancer is defined as the uncontrolled proliferation of mutated tumor forming cells, which have a high ability to metastasize [1]. Despite recent progress made in relation to the early diagnostic and treatment of cancer, a major barrier in terms of being able to successfully eradicate and effectively treat cancer has led to many deaths worldwide. Although chemotherapy and radiotherapy (when utilized singularly or in combination) are current conventional cancer treatments, they are problematic, since these treatment modalities exhibit nonspecific delivery, experience difficulties passing biological barriers, and have low bio-distribution, thus their overall effectiveness is limited and they present additional unwanted side effects [1]. Furthermore, surgery is not considered a complete treatment for some kinds of cancers and it is associated with anaesthesia, high blood loss, and soft-tissue trauma [21]. Therefore, more research is required into unconventional therapies, in order to promote targeted drug delivery and to improve overall treatment outcomes, as well as avoid unnecessary harmful side effects conventional modalities seem to induce.

PDT and PTT are robust, non-invasive, and localized phototherapies, due to their restricted and concentrated treatment area of light exposure [18]. The mechanism of PDT involves the excitation of a photosensitizer (PS) at a specific wavelength [22]. The excited singlet state PS is then promoted to triplet state via intersystem crossing, which consequently transfers its energy to surrounding tissue molecular oxygen and so generates reactive oxygen species (ROS), which obliterates tumor cells via necrotic or apoptotic cell death pathways [22]. PTT is another cancer therapeutic treatment whereby a therapeutic agent is stimulated by both a specific wavelength of light and vibrational energy to produce heat through optical absorption, which consequently kills surrounding cancer cells [23,24]. In PTT, near infrared laser (NIR) is used to irradiate tumor cells topically or interstitially to uniformly increase the temperature in the tumor, as well as prevent damage to healthy cells [23]. When the temperature in the center of tumor reaches a therapeutic temperature above 50 °C, effective ablation occurs [25,26]. However, in some PTT cases this heat may leach out of the target tumor and so damage healthy surrounding cells. In order to circumvent this issue, photothermal absorbers are employed to elevate and concentrate the heat production in target tumor cells only [23]. Furthermore, photothermal absorbers require lowered light energy to reach therapeutic temperature and so only a small amount of heat escapes from the target tissue, preventing any damage to the heathy cells [23]. Various optical dyes such as photosensitizers (PSs) can serve as potential photothermal absorbers, and so can be used to enhance the efficiency of phototherapy. 

Porphyrins and their derivatives are attractive absorbers and are highly conjugated macrocyclic compounds [27] belonging to the four-pyrrole class, which is a vital component of myoglobin and hemoglobin [24]. They are also classed as second-generation PSs, since they have high extinction coefficients and singlet oxygen quantum yields, which are of value in phototherapy cancer treatments [17]. Furthermore, porphyrins are considered as long wavelength-absorbing sensitizers for PDT [28] due to their inherent ability to produce singlet oxygen [17]. They are also biodegradable and ultra-molecular photothermal therapy agents for PTT owing to their high photothermal conversion efficiency and low toxicity [24].

However, since the PDT and PTT therapeutic effect of porphyrins hinge on light energy activation, the application of porphyrin nanomedicines are limited to accessible tumors, such as skin or bladder cancer [29,30]. Thus, researchers tend to incline towards the redshift porphyrin absorption beyond 700 nm, in order to enhance tissue penetration and overall phototherapeutic outcomes [18].

## 3. Phototherapeutic Properties of Porphyrins

Generally, PSs are categorized into porphyrins and nonporphyins [31]. Porphyrins have the following characteristics: they have a low toxicity, high tumor uptake, and stability in the ultraviolet to near infrared region [32]. They absorb light in the therapeutic window wavelength region and so produce ROS through intersystem crossing, allowing them to perform very well within PDT and PTT cancer treatment applications [17]. Their ability to auto-fluoresce within the 400 to 440 nm wavelength region (Soret band) [33] has also made them reliable for PDD applications, such as fluorescence imaging, magnetic resonance imaging (MRI), Raman and photoacoustic imaging (PAI) [34]. Metal free porphyrins show high red fluorescence within 600 to 730 nm, which is the PDT therapeutic range of ideal treatment outcome, since deep penetration of light into tissues can occur [33]. However, metal-free porphyrins often photo-bleach and so their phototherapeutic efficiency within Raman and fluorescence imaging is sometimes limited [17,35].

Metalloporphyrins (porphyrins with metal ions within their inner cavity, such as Gd and Mn [35], are high contrast agents for MRI due to their high absorption coefficient [36]. Zn- and Al- porphyrins have been also employed in the field of PDT [35]. Cationic porphyrins (e.g., zinc(II) meso-tetrakis (N-n-hexylpyridinium-3-yl) porphyrin and 5-(4-trifluoromethylphenyl)-10,15,20-tris(4-trimethylammoniumphenyl) porphyrin) [34] are the most promising PSs for cancer treatment due to their amphiphilic properties [37], since anionic porphyrins such as tetrakis (4-sulfonatophenyl)-porphyrin and their derivatives tend to self-aggregate and so their overall ROS generation is self-limiting [38].

PDT porphyrin-based PSs (PBPs) are further classified into three different generations [31]. First-generation porphyrins (e.g., hematoporphyrin derivative (HpD) and photofrin) showed effective treatment in early clinical trials of PDT on brain [39], colorectal [40], and skin cancer [41]. However, the utilization of the first-generation porphyrins is limited due to their low extinction coefficient, water solubility, chemical impurity, photosensitive toxicity, and suboptimal tissue penetration [17,31]. Therefore, the second-generation porphyrins such as PBPs (e.g., lutrin, purlytin, and tookad) were developed to address the drawbacks of first-generation ones. Despite many significant advantages of second-generation PBPs on melanoma [42], human prostate cancer [43], and breast cancer [44], high dark toxicity, limited cellular uptake and light penetration, as well as severe pain to patients were found [17,31], and so this led the research towards investigating the third-generation porphyrins. Third-generation porphyrins consisting of certain second-generation porphyrin PSs conjugated to various targeting moieties such as antibodies are the current focus of research. Third-generation PSs are basically conjugated to NPs and/or tumor targeting moieties, in order to overcome the limitations of first-and second-generation porphyrins, in relation to subcellular localization and targeted absorption and so overall allow for phototherapeutic efficiency enhancement [45,46,47].

## 4. Cancer Phototherapies with Porphyrin PS-Based Inorganic Nanoparticles

The advent of NPs has allowed for the majority of conventional PS limitations to be overcome. NPs are small in size and range between 1 to 100 nm in diameter and some can contain/carry multiple theranostic agents [48]. Furthermore, their large surface to volume ratios can enhance the payload of PS delivered to target cells [49]. NPs can also prevent premature release of PS, which consequently can prevent nonspecific accumulation in healthy cells [50]. Importantly, NPs give amphilicity to PSs in order to bypass through the blood stream unaffected [51]. Since NPs are small in size when conjugated to PSs, they enhance passive diffusion into tumor cells through the EPR effect [4,52]. Furthermore, PS-NP conjugates can be bound to various targeting ligands to improve active targeted delivery of PSs into deep seated tumors, resulting in a high accumulation of the therapeutic agent and improved PDD or PDT outcomes [4].

Drug delivery systems based on inorganic NPs provide the advantages of a wide surface area conjugation chemistry with versatile surface functionalization [53]. In addition, they bestow great biocompatibility, easy preparation, and excellent physico-chemical properties in order to enhance therapeutic efficacy [53]. It is important to appreciate that non-biodegradable nanoparticles, including inorganic NPs, have recently drawn considerable research interest in the field of PDT, as multifunctional theranostic carriers, due to their unique optical properties and size, shape, and porosity tunability, however is out of the scope of this review. Furthermore, when organic NPs are compared with inorganic polymeric biodegradable NPs, they do not degrade easily into biological systems and do not release a PS efficiently after activation, instead they allow oxygen species to seep out of the carrier. Thus, inorganic NPs (which is the focus of this review) are considered far more superior as they can carry many PSs to the tumor cells as a single nanoplatform, with far more beneficial outcomes [48].

Porphyrins such as protoporphyrin IX (PpIX), purpurin, and benzoporphyrins are the second-generation PSs [17,31,34,35]. These types of porphyrins are important PS agents in cancer phototherapy due to their structural presence of N-H groups and nitrogen atoms, whereby they can be further functionalized by having either nanoparticles, metal ion chelates or polymers conjugated onto their surfaces [54,55]. Since first- and second-generation porphyrins have a high ability to produce ROS in phototherapy cancer applications, they are lipophilic in nature [56], have a low dark toxicity and low water solubility, as well as bio-distribution and their overall emission quantum yields are greatly affected [17,57]. Furthermore, the low passive selectivity of porphyrins for tumor tissue can cause remarkable off-target and unwanted normal tissue damage [58]. Thus, more studies are now focusing on porphyrin-NP conjugated systems for higher phototherapy efficiency, in relation to concentrated tumor cell uptake and so overall improved ROS production for PDT and PTT cancer treatments [48].

### 4.1. Porphyrin-Based Noble Metallic Nanoparticles

Noble metallic NPs such as gold, silver, and platinum have unique properties such as ease of functionalization due to their different chemical moieties and broad optical properties [59]. Additionally, noble metallic NPs can be fine-tuned to desired wavelengths required for porphyrin phototherapies to be effective [59,60]. Furthermore, the localized surface plasmon resonance (LSPR) of gold nanoparticles (AuNPs) [61] and Au metallic components allow for photothermal conversion in cancer phototherapeutic treatments and so contribute to the overall effectivity of the treatment via tumor theranostic heat destruction [62]. However, the low biocompatibility of noble metal NPs sometimes causes restrains within in vivo applications [4].

Within studies performed by Penon et al. (2017), AuNPs conjugates were synthesized using biphasic and monophasic methods to investigate the optimal method of synthesis [63]. These synthesized AuNPs were then conjugated to a porphyrin PS derivative containing an alkanethiol (5-[4-(11-mercaptoundecyloxy) phenyl]-10,15,20-triphenylporphyrin, PR-SH) and a thiolated carboxyl terminated polyethylene glycol (PEG) counterpart [63]. The results from these studies reported that the monophasic method of AuNPs produced the highest amount of singlet oxygen, and so was utilized for PDT treatment assays on in vitro cultured SK-BR-3 human breast cancer [63]. Additionally, the porphyrin-based monophasic AuNPs were also covalently functionalized with a specific breast cancer targeting antibody (Ab) anti-erbB2, to enhance cellular uptake [63]. Overall, a higher cellular uptake was achieved when using the porphyrin-based monophasic-AuNP-PEG-Ab conjugate and a more significant cellular membrane damage was attained after PDT irradiation, when compared to controls [63].

Metalloporphyrins which contain Zinc (Zn) also provide a high PDT efficiency when compared to porphyrins alone, due to their metallic photothermal contribution [64]. However, due to their low solubility and distribution, they are often found conjugated to various vehicles like NPs in order to overcome these shortfalls [65]. Within studies by Alea-Reyes et al. (2017) AuNPs were synthesized with thiol-PEG to promote water solubility and stabilized with amphiphilic gemini-type pyridinium salt [65]. Onto these AuNPs an anionic molecule of Zn(II)meso-tetrakis(4-carboxyphenyl) porphyrin (Zn-TCPP) was incorporated [65]. The synthesized Zn-TCPP loaded AuNPs generated remarkable amounts of singlet oxygen for the PDT treatment of in vitro human breast cancer cell line (SKBR-3), when compared to normal epithelium cells (MCF-10A) [65].

Over the past few years, researchers have shown that the combination of PDT and PTT has higher treatment efficiency than when compared to these treatments in singular form [66,67,68]. In a study, by Zhang et. al. (2019) they integrated synergistic PDT and PTT treatment of in vitro A549 cells and in vivo lung cancer mouse models, using 660 and 808 nm laser irradiations [69]. They fabricated 4-carboxyphenyl porphyrin conjugated silica-coated gold nanorods (AuNR@SiO_2_-TCPP), in which the AuNRs acted as the photothermal conversion agent for PTT, while the TCPP porphyrin PS produced ROS for effective PDT treatment [69]. This study reported that the coating of AuNRs with mesoporous silica, improved PS loading capability and overall drug biocompatibility. In vitro experiments post phototherapy treatment noted a significant cell death of 79% of cultured lung cancer cells [69]. Within in vivo AuNR@SiO_2_-TCPP phototherapy treated mouse models, a remarkable inhibition of tumor volumes was found when compared to control mouse model groups which did not receive treatment. In addition, treated mice with PDT/PTT and AuNR@SiO2-TCPP showed a significant reduction tumor volumes, while AuNRs/PTT and PDT/ AuNR@SiO_2_-TCPP treatments alone produced minimal effect [69].

Since AuNPs exhibit a high tendency to aggregate, researchers have overcome this issue by coating them hydrophilic polymers such as chitosan polyethylene glycol (PEG) or hyaluronic acid in order to promote stability, as well as prevent reduction of heat conversion properties [70]. Additionally, these polymers not only enhance stability and solubility AuNPs, but also reduce their overall biotoxicity [71]. In this regard, studies performed by Zeng et al. (2018) AuNPs were modified with chitosan (QCS-SH) via ligand exchange and then were attached to a PS porphyrin derivative, known as meso-tetrakis (4-sulphonatophenyl) porphyrin (TPPS) for dual PDT and PTT therapy (Figure 1) [58]. This nanohybrid (TPPS/QCS-SH/AuNPs) produced high singlet oxygen and noted high elevated temperature of 56 °C applications when compared to AuNPs or TPPS treatment alone, suggesting that this drug has potential for applications in tumor phototherapy fields [58].

With respect to the combination of PDT and PTT to enhance synergistic phototherapeutic effects, Wei el al. (2018) employed cyclomatrix-type polyphosphazenes to prepare nanosized cross-linked polyphosphazene materials [72]. A tetra(4-hydroxyphenyl) porphyrin (TPP) PS with phenolic hydroxyl groups was then cross linked to this polymer material to form poly(cyclotriphosphazene-co-tetraphenylporphyrin-co-sulfonyldiphenol) nanospheres (CP-TPP) [72]. PEG-AuNPs were then conjugated onto the CP-TPP nanospheres for PDT (630 nm) and PTT (808 nm) in vitro treatment of cervical cancer HeLa cell lines [72]. The abundant presence of nitrogen and phosphorous atoms, as well as phenolic hydroxyl groups on the surface of polyphosphazene materials allow for its successful binding to the surface of AuNPs [73,74]. During characterization experiments the PEG-AuNPs-CP-TPP nanospheres demonstrated high surface plasmon resonance in the infrared (NIR) region for PDT applications and high localized temperature under laser irradiation for PTT assays [75]. Overall, the in vitro cytotoxicity results from this study showed that under sequential irradiation at 630 nm and 808 nm, the cervical cancer cells viability decreased to 10%, when the highest concentration of the drug conjugate was applied and so confirmed the effectiveness of combined PDT and PTT for enhanced cancer therapeutics [72].

Nonselective activation or universal aggregation-caused quenching (ACQ) has greatly decreased the efficiency of conventional PSs for PDT clinical applications [76,77]. ACQ is the main setback in conventional PSs, since they have extended π-conjugation chains, when used at high concentrations and so their fluorescence is reduced PSs remarkably. When a PS aggregates, strong intermolecular π-π stacking interactions occur leading to quenched emission via nonradiative pathways. Thus, conventional PSs can only be utilized at low concentrations, however this affects their photostability [78]. Furthermore, it has been reported that nonselective activation or nonspecific drug leakage of conventional PSs can occur during blood circulation or diffusion into normal tissues which exert unwanted therapy-related side effects, such as toxicity and drug resistance [77].

Furthermore, within clinical trials it had been noted that PDT patients sometimes become photosensitive and so are required to stay away from light to prevent unwanted PS activation until it is completely absorbed by tumor cells, in order to prevent unwanted damage to normal tissues [79]. In this sense, smart PSs have opened a growing research field of PSs. Smart PSs generally remain in the “Off” state during the absorption period and are only selectively activated or turned “On” once they have been fully internalized by cancer tumor cells [80]. Recently, a new phototherapy strategy based on NIR smart PS platforms was proposed by Huang and co-workers (2019) to evaluate its in vitro and in vivo PDT efficiency in breast cancer (4T1) cells and murine tumor induced models [81]. They integrated porphyrin PS units into upper critical solution temperature (UCST) block copolymer decorated gold nanorods (AuNR-P(AAm-co-AN-co-TPP)-b-PEG) [81]. The AuNRs acted as a NIR-manipulated PDT smart PS, as well as a fluorescence quencher of the porphyrin PS and photothermal producer [81]. Results noted that during blood circulation, the UCST block of the copolymer formed a collapsed-core and so caused aggregation of porphyrin PS units, subsequently leading to its “Off” state [81]. Upon internalization of the smart PS nanoplatform into cancer cells and NIR irradiation at 808 nm, the π−π stacking between the porphyrin units broke, activating the PS via a phase transition of UCST polymers from a collapsed to an extended state, causing the porphyrin PS unit to turn “On” (Figure 2) [81]. In this “On” state and localized state within tumor cells, the porphyrin PS then received 650 nm PDT laser irradiation to more effectively generate ROS and singlet oxygen, in order boost the phototherapy efficacy of this smart PDT treatment [81]. Furthermore, the NIR irradiation allowed for a photothermal heating reaction to occur within the AuNRs, which were contained within this drug conjugate and so added to the overall phototherapy efficacy of this smart PDT PS treatment [81]. Overall, this smart PDT “Off/On” state process could be well manipulated using hybrid nanoplatforms with UCST block copolymers and AuNRs, and so could open new prospects for clinical-based PDT treatments [81]. Regarding these in vivo studies, although the tumor growth in the mice injected with the nanoplatforms under 808 and 650 nm laser were remarkably increased, the therapeutic effect was similar to the mice treated with the nanoplatform under 650 nm laser only [81].

Generally, phototherapeutic agents are mainly locate in adjacent regions of a tumor, due to their abnormal vasculature nature and high interstitial fluid pressure (IFP), thus large parts of the tumor remain unaffected, since they do not adequately absorb PSs [82]. However, evidence has shown that many types of important cells associated with tumor initiation and progression are those fed by the defective blood vessels which supply solid tumors [83,84] and research must focus on phototherapy research to target these cells, which in turn can prevent tumor growth and metastases [85]. Various strategies have focused on adjacent cells, while those distant from blood vessels remain untouched [86]. Therefore, the development of novel strategies to improve the penetration and uptake properties of PSs still remains an obstacle in nanotherapeutics.

Two-photon PDT (2P-PDT) is a promising treatment approach that uses the absorption of two photons with low energy in NIR region (650–950 nm) to provide deeper penetration, as well as a accurately target cancer tissues [87]. Some conventional PSs tend to have a week emission and lack of imaging possibilities and 2P-PDT can improve these limitations since this modality consists of small two-photon absorption (2PA) cross sections [87,88].

Inspired by 2P-PDT, Li et al. (2019) successfully conjugated poly [9,9′-bis(6″-bromohexyl) fluorene-2,7-ylenevinylene-co-alt-1,4-phenylene] (PFV) as a two-photon light harvesting material to tetraphenylporphyrin (TPP) to improve its PDT efficiency [89]. Conjugated polymer nanoparticles (CPNs) were then covalently attached to silica coated AuNRs (AuNR@SiO_2_–CPNs) [89]. Fluorescence resonance energy transfer from PFV and plasmon resonance through AuNRs enhanced the two-photon properties of the TPP [89]. The nanohybrid was further applied for simultaneous two-photon imaging and 2P-PDT in vitro treatment of Hela cells which revealed significant enhancement in two-photon excitation fluorescence (980-fold) and two-photon induced singlet oxygen generation efficiency (792-fold) to eradicate cervical cancer cells [89].

A new treatment modality known as sonodynamic therapy (SDT) was proposed by Liang et al. (2019). SDT is used to excite and so activate PSs to produce high enough ROS levels for effective cancer therapy [90]. Furthermore, as already mentioned, photothermal absorbers or photothermal transduction agents (PTAs) are adopted in PTT to harvest light energy and generate hyperthermia [91]. These PTAs are categorized into semiconductor nanocrystals (NCs) [92], inorganic materials [93], and organic dyes [94]. It has been reported that platinum or palladium-based photothermal transduction agents (PTAs) as inorganic PTAs have a higher photothermal stability and better catalytic properties than when compared to Au-based PTAs [95,96]. Furthermore, studies have noted that copper sulfide semiconductor NCs provide very high LSPR within the NIR region [97].

In this regard, studies developed platinum-copper sulfide Janus nanoparticles conjugated to tetra-(4-aminophenyl) porphyrin (TAPP) to overcome the low penetration depth of PDT [90]. In fact, they integrated semiconductor NCs and noble platinum metal to form their PTAs [90]. The noble platinum metal promoted the photothermal conversion efficiency under 808 nm laser irradiation by changing the electron transport pathway [98] and the large space of the hollow copper sulfide NPs interior facilitated a high loading capacity of the TAPP PSs [90]. The synthesized nanohybrid was further coated with a temperature-sensitive polymer consisting of (poly (oligo (ethylene oxide) methacrylate-co-2-(2-methoxyethoxy) ethyl methacrylate) to increase the biocompatibility and temperature triggered drug controlled release [90]. Therefore, upon laser irradiation and catalysis-enhanced SDT, this PS nanocarrier produced hyperthermia and a large amount of singlet oxygen which successfully managed to eradicate CT26 murine colon cancer cells within in vitro experiments with a cell viability of 19.8%. The in vivo studies using the CT26 xenograft tumor model in the group treated with the nanohybrid under laser and ultrasound irradiation demonstrated complete removal of the tumor without reoccurrence [90].

Although AuNPs have a two distinctive absorption bands within the visible region, one on the short axis (transverse) and long axis (longitudinal) [99], enabling their ease of detection, this does cause them to exhibit low quantum yields [100]. So research has inclined towards engineering such NPs into nanocubes in order to heighten quantum yields and so allow for biological tissue detection [100,101]. Furthermore, AuNPs can not only attenuate X-rays for molecular imaging [102], but can be also used in photoacoustic imaging via thermo-elastic expansion, which can be distinguished or identified by ultrasound detectors [103,104].

### 4.2. Porphyrin-Based Magnetic Nanoparticles

Magnetic nanoparticles are ideal contrast agents applied in MRI [105]. In addition, their large surface to volume ratio and biocompatibility facilitate conjugation of targeting ligands and therapeutic drugs onto their surfaces [105]. The main limitations of magnetic nanoparticles are intercellular aggregation and low water solubility; however this can be overcome by encapsulation of hydrophilic polymers onto their surfaces [106].

A multifunctional nanosystem known as Fe_3_O_4_@mSiO_2_ (TPPS)-PL was fabricated by Liu et al. (2017) for the in vitro PDT treatment of HepG2 and HeLa cells [107]. The Fe_3_O_4_ magnetic NPs were further coated with mesoporous silica and Pullulan (PL) targeting moieties were covalently attached onto the Fe_3_O_4_@mSiO_2_. Tetraphenylporphyrin tetrasulfonic acid hydrate (TPPS) was then loaded into the silica shell of the nanocomposite to form Fe_3_O_4_@mSiO_2_ (TPPS)-PL NPs [107]. PL is a water-soluble, nonionic polysaccharide [108] which has a low toxicity and immunogenicity [107]. PL binds to the asialoglycoprotein receptor [109] which is overexpressed on HepG2 cells with maximum receptor activity [110]. The cellular uptake results from this study showed that the NPs internalized far better in HepG2 cells than Hela cells, due to the high asialoglycoprotein receptor overexpression in HepG2 cells, confirming the effective targeting ability of this nanocomposite [107]. Within PDT in vitro experiments with 200 µg mL^−1^ Fe_3_O_4_@mSiO_2_ (TPPS)-PL NPs caused 65% loss of cell viability in HepG2 cells, while 100 g mL^−1^ of the nanocomposite noted only 20% of cells remained viable, after 20 min of treatment, suggesting the overall efficacy of this treatment for human hepatoma carcinoma [107].

Thus, the high biocompatibility and biodegradability of magnetic NPs are powerful properties within biomedical applications [111]. Although magnetic NPs like iron oxide NPs are being used in MRI due to their high negative contrast, the detection of negative contrast agents in tissues with low signal intensity, such as lungs and blood clots, is challenging, making cancer diagnosis sometimes difficult [1]. In this sense, positive contrast NPs such as manganese oxide NPs are ideal candidates for low signal intensity cancerous tissues [112].

### 4.3. Porphyrin-Based Carbon Nanoparticles

Various nanocarbons including carbon nanotubes (CNTs), fullerene, nanodiamonds, and carbon nanoparticles (CNPs) are potential assets in phototherapeutic applications [113]. Their tunable surface structure, high drug loading capacity and noticeable fluorescence are promising features of nanocarbon NPs and tubes as PS nanocarriers and theranostic contributors to PDT [4].

Various studies have noted that metal and nitrogen co-doped carbon nanospheres could produce porphyrin-like metal coordination centers, with the ability to change redox states and electronic structures to form active sites [114,115]. In this regard, Chen et al. (2018) synthesized gadolinium encapsulated graphene CNPs (Gd@GCNs) [116]. They suggested that this synthesized nanoplatform provided a strong spin-orbit coupling at the heavy metal center resulting in significant intersystem crossing and ROS generation under photoirradiation [116]. The mechanism of action of this synthesized nanoplatform has also been proposed in other studies investigating the doping of metal into the core of CNPs [117,118,119]. The synthesized Gd@GCNs showed high singlet oxygen quantum yields and strong fluorescence when applied in dual-modal fluorescence imaging and MRI [116]. This nanocomposite also demonstrated satisfactory image-guided PDT outcomes for the in vivo cancer treatment of SCC-7 tumor induced mouse models. Furthermore, the tumor growth significantly delayed in the mice injected with Gd@GCNs. A tumor inhibition rate of 2.4 ± 1.6 was also observed in the PDT group on day 12 (Figure 3) [116].

The high surface area and loading capacity, together with the release of vibrational energy induced by NIR exposure are ideal properties CNTs present for theranostic applications [120,121]. However, the low solubility and toxicity of CNTs still require ongoing investigations within clinical tests [122].

### 4.4. Porphyrin-Based Silica Nanoparticles

The main distinguished features of silica NPs is their distinct nanostructure and well-established siloxane chemistry, which facilitates the use of various methods of surface modifications for diagnostic and therapeutic applications [123,124]. They also enable the fabrication of different silica nanoplatforms with various morphologies, sizes and porosity such as hollow or mesoporous silica NPs [125].

Martínez-Carmona et al. (2017) proposed a visible light-responsive nano-drug delivery which comprised of silica NPs (MSN) decorated with porphyrin-caps to deliver topotecan (TOP) [126]. The authors noted higher tissue penetration of visible light when compared to UV light for the PDT treatment of in vitro HOS osteosarcoma cancer cells. In the presence of visible light, the porphyrin-nanocaps produce singlet oxygen molecules which broke the sensitive-linker and triggered pore uncapping, allowing the release of the entrapped TOP [126]. This nano-drug system was non-toxic and the greater penetration capacity of visible radiation noted a double antitumor effect due TOP release and porphyrin ROS production [126]. Furthermore, in vitro assays revealed that TOP was released in controlled fashion inside HOS osteosarcoma cancer cells, via drug endosome escape to reach the cytoplasm [126,127]. This research work opened up promising expectations for new alternative drug delivery systems for cancer treatment [126].

Chemotherapeutic drugs such as gemcitabine hydrochloride are noted to have a short in vivo half-life and poor membrane permeability due to its hydrophilic nature [128]. Thus, high amounts of this drug is required to be administered to patients in order to effectively eradicate cancer tumor cells, however this induces adverse unwanted side effects [128]. In this sense, a nanodelivery system was proposed by Aggad and co-workers (2018) for gemcitabine cancer therapy in order to overcome the hydrophilic limitations of chemotherapeutic drugs [128]. In this study, they synthesized ethylene-based periodic mesoporous organosilica NPs (PMOs) for 2P-PDT and in order to enhance the delivery of gemcitabine within in vitro cultured MCF-7 breast cancer cells [128]. A tetrasilylated porphyrin (PS1) PS was then attached to the ethylene-based PMOs, which caused J-aggregation inside the meso-structure of NPs leading to a two-photon PDT effect [128]. Generally, PSs aggregate with absorption bands shifted to a longer wavelength, than when compared to monomer bands known as J-aggregates, which enhance the two-photon absorption properties [129,130]. The synergistic effect of the two-photon irradiation with gemcitabine delivery and PS1-EPMOs noted more significant cancer cell death than when compared to control cells which did not receive irradiation [128].

Ultra-small hollow silica nanocarriers (HSdots) (~10 nm) were fabricated within studies performed by Dam et al. (2019) [131]. These HSdots were then coated with a porphyrin zinc phthalocyanine (ZnPC) PS for the in vitro PDT treatment of human cutaneous and head/neck squamous cell carcinomas (SCC) cell lines-A431, SCC12 and CAL27, with normal human epidermal keratinocytes (NHEK) as a normal control cells line [131]. Studies have reported that silica nanomaterials are highly biodegradable and so extremely suitable for drug delivery in cancer cells [132]. A higher cellular uptake of ZnPC-loaded HSdots was reported in SCC cell lines, due its cancer targeted modification with folic acid, in comparison to NHEK cells (Figure 4A–D) [131]. Furthermore, the in vitro PDT treatment of SCC cells at 671 nm laser exposure resulted in more than 90% cell death, whereas no evidence of dead cells was found in NHEK cells (Figure 4E), suggesting this form of treatment for skin cancer is highly probable [131].

In order to increase the lifetime of nanocarriers in the bloodstream and to delay opsonization and phagocytosis of the NPs, they are often grafted into a protective hydrophilic layer of polysaccharides, such as xylan or long chain polymers of PEG [133,134]. According to studies by Bretin et al. (2019), silica nanoparticles (SNPs) were successfully coated with xylan for the enhanced drug delivery of a 5-(4-hydroxyphenyl)-10,15,20-triphenylporphyrin (TPPOH) PS for the in vitro PDT treatment of human colorectal cancer cells (CRC) [135]. Results showed that vectorizing xylan coated SNPs with TPPOH (TPPOH-X SNPs) not only improved anticancer efficacy of PSs PDT in vitro CRC induced apoptotic cell death, also enhanced the in vivo tumor cytotoxicity in CRC xenografted mouse models. In vivo treated mice showed increased apoptosis and cell proliferation inhibition [135].

Multi-modal synergistic phototherapeutic functionalities with NPs have recently been explored by Li et al. (2020) for the treatment of human breast carcinoma (MCF-7) and human lung cancer (A549) in vitro cultured cell lines [136]. Within this study researchers integrated silicon nanoparticles (SiNPs) with 5,10,15,20-tetrakis (1-methyl 4-pyridinio) porphyrin tetra (p-toluenesulfonate) (TMPyP) PS and the anticancer drug known as doxorubicin (DOX), to generate singlet oxygen for effective PDT under light irradiation, with chemotherapeutic benefits, respectively [136]. Additionally, due to the two-photon excitation fluorescence abilities of SiNPs, researchers could produce cellular imaging at the NIR laser excitation [136]. Furthermore, Folic acid (FA) was also conjugated to the nanocomposite (MSN@SiNPs@TMPyP-FA) for enhanced targeting capabilities of cancer cells (Figure 5) [136]. Significant blue two-photon fluorescence of the SiNPs from MSN@SiNPs@TMPyP-FA was observed within the cytoplasm of MCF-7 cells, which have more overexpressed FA receptors, in comparison to A549 cells which noted a weaker fluorescence signal, due to their lowered expression of FA receptors [136]. Within MCF-7 cells which received MSN@SiNPs@TMPyP-FA and PDT only 30% viability was noted, when compared to monotherapy controls [136].

Within an in vivo treatment study performed on 4T1 murine breast tumor-bearing mice by Goel et al. (2018), various breast cancer diagnosis and phototherapies such as; PDT-660 nm/PTT-980 nm treatments, positron emission tomography (PET), fluorescence (FL), cerenkov luminescence (CL), and cerenkov radiation energy transfer-based imaging (CRET) were integrated [137]. This study fabricated a multifunctional core-satellite nanoconstruct (CSNC), by coordinating abundant copper sulfide NPs on the surface of hollow mesoporous silica nanoshells (HMSN) decorated with radiolabel [^89^Zr] [137]. Meso-tetrakis (4-carboxyphenyl) porphyrin (TCPP) was then conjugated onto the nanohybrid to form a highly efficient multimodal nanotheranostic cancer diagnostic and treatment drug [137]. Overall this multimodal therapy is revolutionary, since it demonstrated the foresight of diagnosis, reported no toxicity and show a significant volume reduction of murine breast tumors without reoccurrence after 30 days [137].

### 4.5. Porphyrin-Based Upconversion Nanoparticles

Rare earth-based upconversion nanoparticles (UCNPs) have emerged recently in research as a way to circumvent the low tissue penetration depth limitation of PSs [138,139]. Generally, the conversion of NIR light to a shorter wavelength of light is known as upconversion, which is an anti-Stokes process [140,141]. Upconversion luminescence is a nonlinear process whereby successive lower energy photons absorb luminescence and so emit higher energy photons [140,142]. This process has the advantage of low light scattering and autofluorescence background, as well as high tissue penetration depth, since this excitation occurs in the NIR region which is located within phototherapeutic window. This also consequently decreases any photo damage biological tissues might experience [143].

In a study by Sun et al. (2019), protoporphyrin IX PS was modified with jeffamine (PJ) to improve its hydrophilicity and biocompatibility [144]. PEGylated UCNPs were then covalently conjugated to protoporphyrin IX-PJ to control the resonance energy transfer from the UCNPs to the PJs [144]. The authors proposed that the UCNPs-PJ had a tumor targeting ability, due to the presence of low density lipoprotein (LDL) receptors on the surfaces of cancer cells, which enables the passive capture protoporphyrin IX-PJ-UCNPs [144]. Therefore, the study assessed the PDT treatment abilities of this UCNP composite at 980 nm laser irradiation on an in vitro cultured intestinal human colon adenocarcinoma cell line (LS180) cell line, which is known to over-express LDL receptors [144]. Results reported improved uptake of the PS when conjugated onto the UCNP composite, with significant eradication of cancer cells [144].

A novel caspase-3 responsive functionalized UCNP was fabricated for multi-functional modalities of NIR triggered PDT damage, caspase-3 activation, caspase-3 responsive drug release, and cascade chemotherapeutic activation of DOX by Zhao et al. (2017) [145]. The synthesized UCNPs comprised of pyropheophorbide-a methyl ester (MPPa) PS, tumor-targeting cRGD-PEG-DSPE, and caspase-3 responsive DOX prodrug tethered with DEVD peptide (DEVD-DOX) forming MPPa/UCNP-DEVD-DOX/Crgd [145]. Upon NIR irradiation, energy from the UCNPs was transferred to the PS which generated ROS for PDT treatment and simultaneously activated caspase-3 to initiate apoptotic cell death (Figure 6) [145]. The cleavage of the peptide sequence within DEVD-DOX occurred due to the activated caspase-3 enzyme resulting in DOX release for chemotherapy treatment [145]. The in vitro NIR PDT treatment of 4T1 murine breast cancer cells with this multi-functional UCNP demonstrated significantly high cytotoxicity, when compared to control cells which did not receive NIR irradiation [145]. Moreover, in vivo treatment of 4T1 tumor bearing mice with the cascade nanotherapeutic revealed high numbers of pyknotic cells with condensed nuclei, which were either undergoing apoptotic cell death or were completely disintegrated [145].

Within PDT applied UCNPs, multiple low-energy exciting photons are used to emit a higher energy photon instead of the excitation of a single photon alone [138]. Therefore, UCNPs can achieve improved sensitivity with a low autofluorescence [138]. Furthermore, since UCNP PSs are activated with NIR, the PDT light penetration depth is increased, since NIR can achieve a skin penetration depth of up to 3 mm [138]. Additionally, UCNP can be excited/activated by X-rays and so are ideal candidates for photodiagnostic detection of deep-seated tumors [1].

### 4.6. Porphyrin-Based Quantum Dots

Quantum dots (QDs) are small nanocrystals which range from 2 to 10 nm in size, as well as have unique chemical and physical properties [146]. The surface of the QDs can be modified with thiol ligands or amphiphilic copolymers to facilitate bioconjugation of antibodies or small drugs, to improve the solubility and specificity of tumor targeting porphyrin-based PS delivery [146].

Cao et al. (2017) synthesized a porphyrin derivate (P) that reported a high singlet oxygen quantum yield and then conjugated it to PEG and aptamer-functionalized graphene QDs (GQDs), forming multifunctional photodiagnostic and theranostic agent (GQD-PEG-P) for cancer treatment [147]. This theranostic nanoplatform was investigated for its PDT/PTT synergic therapy within in vitro cultured A549 lung cancer cells [147]. The high fluorescence signal produced by the GQDs allowed for differentiation between A549 lung cancer cells and somatic normal cells and due to their large surface area, it enabled the detection of intracellular cancer-related microRNA (miRNA) in cancer cells [147]. The in vitro combined treatment of A549 cells with PDT/PTT under 635 nm and 980 nm laser irradiation could obliterated cancer cells more effectively, when compared to individual treatment modalities administered singularly [147]. Furthermore, MCF-7 breast cancer multicellular tumor spheroids (MCTS) were also assessed for the in vivo GQD-PEG-P PDT and PTT phototherapeutic effects and results noted that most cells were dead after treatment [147].

Despite the great properties QDs possess in terms of high tunability with high quantum yields [1], there are limited studies regarding their application in porphyrin-based PDT cancer treatments. The reason behind the limitation of QD porphyrin-based PDT applications is due to the high toxicity they possess, since most consist of toxic heavy metals like cadmium ions [1]. The application of cadmium free QDs such as zinc and indium-based QDs [148] or the conjugation of QDs to the surface or incorporation into polymeric NPs [149] are a promising area of research, since this enhances their overall application in clinical trials in order to investigate their overall effectiveness for cancer diagnosis and treatment [150]. However, due to the limited penetration of visible light, the utilization of both Cd containing and Cd-free QDs is restrained to superficial tumors for diagnosis of cancer only [1,151]. Nevertheless, QDs possess a high two-photon absorption (TPA) cross-section (σ~10^4^ GM) [152], which can be employed in two-photon bioimaging [153]. Furthermore, polymer-encapsulated QDs (P-QDs) have shown a promising biocompatibility improvement, with minimal cytotoxicity in cells and animals, when compared to standard QDs [153,154].

## 5. Conclusions and Perspectives

Cancer is a dreaded disease causing a vast number of deaths worldwide. Despite ongoing, in-depth research into ways of improving cancer treatment, novel treatment modalities, which obliterate cancer specific tumors, without affecting normal tissues, is still in high demand. The arrival of nanotechnology and engineered nanomedicines have provided a novel opportunity to try and diagnose, as well as treat cancer, by promoting enhanced drug delivery.

Phototherapy is a relatively new unconventional treatment modality that is being investigated for cancer treatment, since it exhibits limited side effects. This review highlights the recent investigations of porphyrin-based nanomedicines for their phototherapeutic and diagnostic applications for cancer over the last 3 years (2017 to 2020). The focus of this review was on the inorganic-based nanoparticles incorporated with porphyrin PSs for phototherapy treatment of cancer.

Various inorganic NPs have been scrutinized for porphyrin delivery. Former porphyrin loaded inorganic NPs in cancer treatment were evaluated by Zhou et al. (2016) [155] and studies by Xue et al. (2019) thoroughly reviewed porphyrin-based organic NPs for cancer therapy [18]. Therefore, this review serves as a current update from these previous reviews, however with a core focus on inorganic porphyrin-based nanomedicines within PDD, PTT, and PDT application for cancer research.

Porphyrin PSs are the next generation of PSs which can be conjugated to various NP moieties, for improved multiple phototherapy and diagnostic functions in relation to cancer treatment regimes. However, despite its merits, the greatest drawback for porphyrin-based nanomedicines remain the limited penetration of light to deep seated tumors. This remains the greatest challenge when using porphyrin-based nanomedicines for PDT and PTT. Thus, porphyrin-inorganic NP-based PSs, for now, can only be applied for light accessible tumors, such as skin or bladder cancer [19]. However, recent studies have reported the use of UCNPs, 2P-PDT, and X-rays for cancer phototherapy, where inorganic porphyrin-based NPs can absorb energy at much higher wavelengths, releasing a larger fluorescence signal, as well as allow for a higher wavelength of light which improves penetration of light in tissue and so largely expands their therapeutic and diagnostic applications [156,157,158]. Furthermore, the treatment of cancer metastases is also challenging with porphyrin-based nanomedicines, due to the lack of specific cellular localization. To tackle this limitation, phototherapy using porphyrin-based nanomedicines is often integrated with immunotherapy or targeting ligands and antibodies, which enhance the active uptake of PSs via overexpressed cancer cell receptors [159,160,161]. Lastly, it is important to note that more clinical based in vivo cancer research studies need to be performed in order to completely understand the pharmacological benefits porphyrin-based nanomedicines have [17].

Future directives using inorganic porphyrin NP include the use of PSs in combination with other chemotherapeutic drugs, as well as developing specific targeting nanoplatforms to boost the therapeutic uptake of drugs, limiting off-target uptake to prevent detrimental side effects and so enhancing cancer treatment.

## Figures and Tables

**Figure 1 ijms-21-03358-f001:**
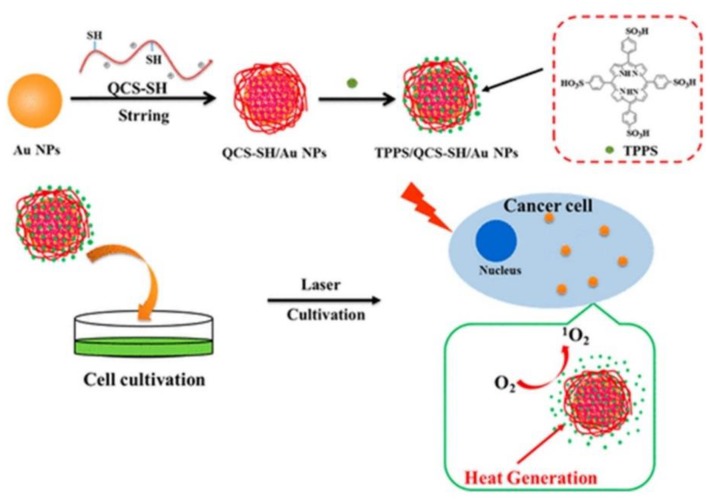
Proposed fabrication of meso-tetrakis (4-sulphonatophenyl) porphyrin (TPPS)/QCS-SH/gold nanoparticles (AuNPs) for dual mode photodynamic therapy (PDT)/photothermal therapy (PTT) treatment of cancer. Reprinted with permission from ref. [58]. Copyright 2018 American Chemical Society.

**Figure 2 ijms-21-03358-f002:**
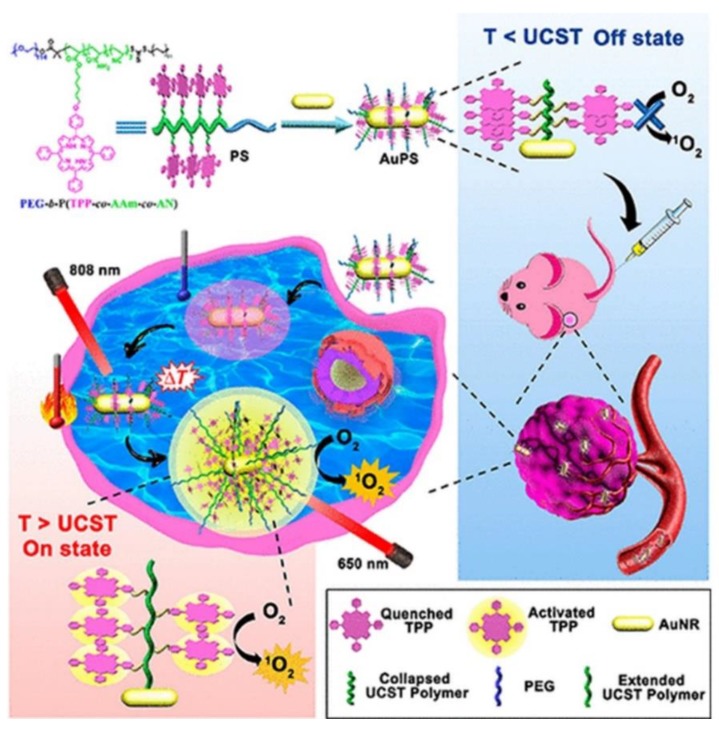
Schematic preparation of the “Off/On” upper critical solution temperature (UCST) smart nanoplatform process for the PDT clinical treatment in murine breast cancer models. Reprinted with permission from ref. [81]. Copyright 2019 American Chemical Society.

**Figure 3 ijms-21-03358-f003:**
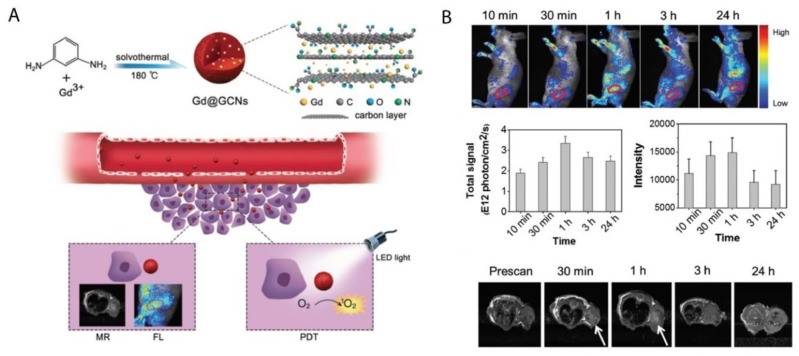
(**A**) Preparation of Gd@GCNs nanoplatform for imaging-guided phototherapy in solid tumors, (**B**) In vivo fluorescence images acquired from tumor-bearing nude mice after intravenous injection of Gd@GCNs at different time intervals, noting significant tumor shrinkage over time after PDT treatment. Reprinted with permission from ref. [116]. Copyright 2018 Wiley Online Library.

**Figure 4 ijms-21-03358-f004:**
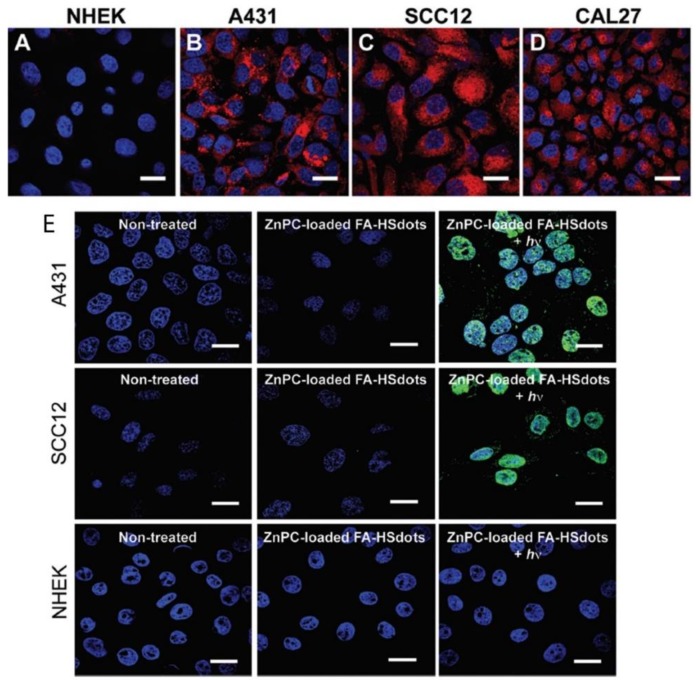
Cellular uptake of ZnPC-loaded FA–HSdots labeled with TRITC in (**A**) normal human epidermal keratinocytes (NHEK) and human cutaneous and head/neck (SCC) lines (**B**) A431, (**C**) SCC12, (**D**) CAL27 and (**E**) indicates the PDT therapeutic efficacy of ZnPC-loaded FA–HSdots observed in all SCC cells and control NHEK cells after treatment with 2 min laser exposure. Reprinted with permission from ref. [131]. Copyright 2019 Royal Society of Chemistry.

**Figure 5 ijms-21-03358-f005:**
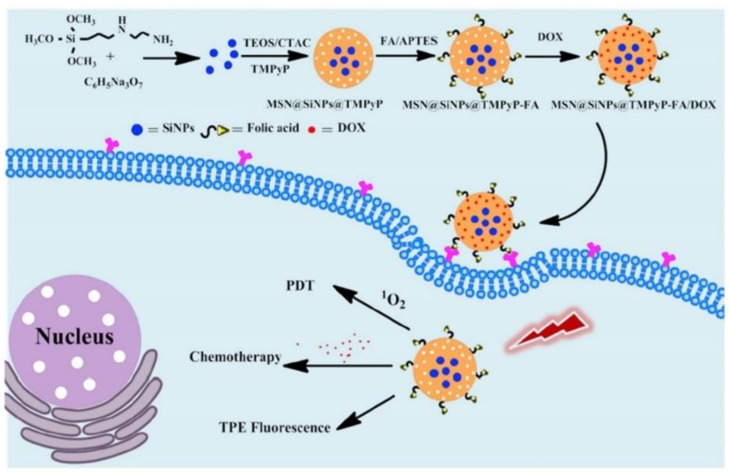
Schematic illustration of the preparation of MSN@SiNPs@TMPyP-FA for enhanced PS and doxorubicin (DOX) drug delivery for improved PDT and chemotherapeutic outcome treatments in cancer cells, with TPE fluorescence. Reprinted with permission from ref. [136]. Copyright 2020 Elsevier.

**Figure 6 ijms-21-03358-f006:**
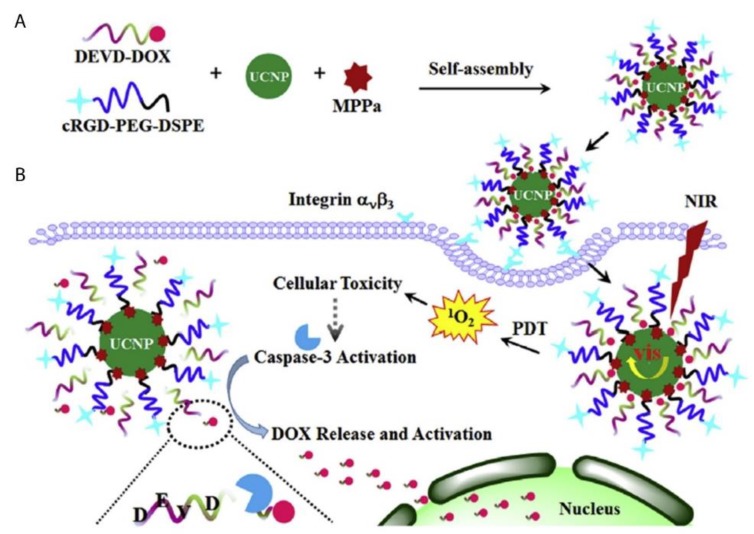
Schematic illustration of (**A**) fabrication of functionalized upconversion nanoparticles (MPPa/UCNP-DEVD-DOX/cRGD), (**B**) proposed mechanism of PDT action, activation of the caspase-3 and release of chemotherapeutic DOX. Reprinted with permission from [145]. Copyright 2017 Elsevier.

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
