# Peer review of "Recent Advances in Porphyrin-Based Inorganic Nanoparticles for Cancer Treatment"

_ijms, 2020, doi:10.3390/ijms21093358_

Round 1
Reviewer 1 Report
This is a potentially interesting review on porphyrins for PDT/PTT and combinations thereof. It requires substantial editing.
An overall copy edit is required for grammar and word usage. There are many instances of subject/verb disagreement and strange word use, for example “In this sense” to mean “for this reason.”
Line 39—PTT is described too briefly. Where does the heat energy come from (vibrations?). What types of photosensitizers are typically used?
For the connection to the next paragraph: are porphyrins useful for both PTT and PDT? Please clarify.
L 75 : Their ability to 74 auto-fluoresce within 400 to 450 nm wavelength region
The Soret band is ~400 nm. What is Stokes shift? It would be helpful to show absorbance/emission peaks.
L82: “Porphyrins are categorized into three different classes” Based upon what?
L174: “Nonselective activation or universal aggregation-caused quenching (ACQ) has greatly decreased the efficiency of conventional PSs for…” These are very different issues and should be discussed separately.
L204: “Two-photon PDT (2P-PDT) is a promising treatment approach that uses the absorption of two 204 photons with low energy in NIR region (650-950 nm) to provide deeper penetration..” this is a sudden shift from the previous paragraph. The importance of greater penetration has not been discussed.
L 235: “Although AuNPs have a high optical activity within the visible region, enabling their ease of detection, they often tend to exhibit low quantum yields”—are we discussing absorbance or emission here? Most, except the smallest, are not fluorescent. Please clarify.
L 436: “PDT research generally restrains from the utilization of cadmium free QDs” This section isn’t entirely clear. It is clear that Cd is toxic and that non-Cd-containing QDs are needed. But this statement: “due to their limited penetration of visible light for superficial tumours” should apply to both Cd-containing and Cd-free QDs. If not, please clarify. What about 2-photon excitation?
Author Response
Reviewer #1:
This is a potentially interesting review on porphyrins for PDT/PTT and combinations thereof. It requires substantial editing.
An overall copy edit is required for grammar and word usage. There are many instances of subject/verb disagreement and strange word use, for example “In this sense” to mean “for this reason.”
- Manuscript has been reviewed by an English Language expert and been improved upon.
Line 39—PTT is described too briefly. Where does the heat energy come from (vibrations?).
PTT mechanism has been moved to Section 2 and the following paragraph additions and references have been added within this Section:
- PTT is another cancer therapeutic treatment whereby a therapeutic agent is stimulated by both a specific wavelength of light and vibrational energy to produce heat through optical absorption, which consequently kills surrounding cancer cells [23, 24]. In PTT, near infrared laser (NIR) is used to irradiate tumour cells topically or interstitially to uniformly increase the temperature in the tumour, as well as prevent damage to healthy cells [23]. When the temperature in the centre of tumour reaches a therapeutic temperature above 50°C, effective ablation occurs [25, 26]. However, in some PTT cases this heat may leach out of the target tumour and so damage healthy surrounding cells. In order to circumvent this issue, photothermal absorbers are employed to elevate and concentrate the heat production in target tumour cells only [23]. Furthermore, photothermal absorbers require lowered light energy to reach therapeutic temperature and so only a small amount of heat escapes from the target tissue, preventing any damage to the heathy cells [23]. Various optical dyes such as photosensitizers (PSs) are can serve as potential photothermal absorbers, and so can be used to enhance the efficiency of phototherapy.
What types of photosensitizers are typically used?
This statement has been modified to clarify in the introduction:
- Amongst the different generations of PS available today, current research seems to be focus on the development or investigation into various porphyrin PS types. Porphyrin PSs and their derivatives are organic heterocyclic macrocycles with a high phototoxicity [16] can be applied in multiple diagnostic and therapeutic functions.
For the connection to the next paragraph: are porphyrins useful for both PTT and PDT? Please clarify.
The following paragraph has been added to Section 2 to clarify the use of porphyrin for PDT and PTT:
- Porphyrins and their derivatives are attractive absorbers and are highly conjugated macrocyclic compounds [27] belonging to the four-pyrrole class, which is a vital component of myoglobin and haemoglobin [24]. They are also classed as second-generation PSs, since they have high extinction coefficients and singlet oxygen quantum yields, which are of value in phototherapy cancer treatments [17]. Furthermore, porphyrins are considered as long wavelength-absorbing sensitizers for PDT [28] due to their inherent ability to produce singlet oxygen [17]. They are also biodegradable and ultra‑molecular photothermal therapy agents for PTT owing to their high photothermal conversion efficiency and low toxicity [24].
L 75: Their ability to auto-fluoresce within 400 to 450 nm wavelength region. The Soret band is ~400 nm. What is Stokes shift? It would be helpful to show absorbance/emission peaks.
The following paragraph additions and references have been added in Section 3 to clarify the absorption/emission peaks of porphyrins:
- Their ability to auto-fluoresce within the 400 to 440 nm wavelength region (Soret band) [33], has also made them reliable for PDD applications, such as fluorescence imaging, magnetic resonance imaging (MRI), Raman and photoacoustic imaging (PAI) [34]. Metal free porphyrins show high red fluorescence within 600 to 730 nm, which is the PDT therapeutic range of ideal treatment outcome, since deep penetration of light into tissues can occur [33].
L82: “Porphyrins are categorized into three different classes”. Based upon what?
This sentence has been modified in Section 3 for further clarity:
- PDT porphyrin based PSs (PBPs) are further classified into three different generations [31].
L174: “Nonselective activation or universal aggregation caused quenching (ACQ) has greatly decreased the efficiency of conventional PSs for…” These are very different issues and should be discussed separately.
The following paragraph additions and references have been included in Section 4.1 for clarification:
- ACQ is the main setback in conventional PSs, since they have extended π-conjugation chains, when used at high concentrations and so their fluorescence is reduced PSs remarkably. When a PS aggregates, strong intermolecular π-π stacking interactions occur leading to quenched emission via nonradiative pathways. Thus, conventional PSs can only be utilized at low concentrations, however this affects their photostability [78]. Furthermore, it has been reported that nonselective activation or nonspecific drug leakage of conventional PSs can occur during blood circulation or diffusion into normal tissues which exert unwanted therapy-related side effects, such as toxicity and drug resistance [77].
L204: “Two-photon PDT (2P-PDT) is a promising treatment approach that uses the absorption of two photons with low energy in NIR region (650-950 nm) to provide deeper penetration..” this is a sudden shift from the previous paragraph. The importance of greater penetration has not been discussed.
The following paragraph and references have been added to the manuscript before 2P-PDT in Section 4.1 to discuss the importance of deep penetration:
- Generally, phototherapeutic agents are mainly locate in adjacent regions of a tumour, due to their abnormal vasculature nature and high interstitial fluid pressure (IFP), thus large parts of tumour remain unaffected, since they don’t adequately absorb PSs [82]. However, evidence has shown that many types of important cells associated with tumour initiation and progression are those fed by the defective blood vessels which supply solid tumors [83, 84] and research must focus on phototherapy research to target these cells, which in turn can prevent tumor growth and metastases [85]. Various strategies have focused on adjacent cells, while those distant from blood vessels remain untouched [86]. Therefore, the development of novel strategies to improve the penetration and uptake properties of PSs still remains an obstacle in nanotherapeutics.
L 235: “Although AuNPs have a high optical activity within the visible region, enabling their ease of detection, they often tend to exhibit low quantum yields”—are we discussing absorbance unless otherwise stated or emission here? Most, except the smallest, are not fluorescent. Please clarify.
This sentence has been modified in Section 4.1 for clarity:
- Although AuNPs have a two distinctive absorption bands within the visible region, one on the short axis (transverse) and long axis (longitudinal) [99], enabling their ease of detection, this does cause them to exhibit low quantum yields [100].
L 436: “PDT research generally restrains from the utilization of cadmium free QDs” This Section isn’t entirely clear.
It is clear that Cd is toxic and that non-Cd-containing QDs are needed. But this statement: “due to their limited penetration of visible light for superficial tumours” should apply to both Cd containing and Cd-free QDs. If not, please clarify.
This sentence has been modified in Section 4.6 for clarification:
- The application of cadmium free QDs such as zinc and indium based QDs [148] or the conjugation of QDs to the surface or incorporation into polymeric NPs [149] are a promising area of research, since this enhances their overall application in clinical trials in order to investigate their overall effectiveness for cancer diagnosis and treatment [150]. However, due to the limited penetration of visible light, the utilization of both Cd containing and Cd-free QDs is restrained to superficial tumours for diagnosis of cancer only [1, 151].
What about 2-photon excitation?
Regarding 2-photon excitation, the following paragraph and references have been added:
- Nevertheless, QDs possess a high two-photon absorption (TPA) cross-Section (σ~104 GM) [152], which can be employed in two-photon bioimaging [153]. Furthermore, polymer-encapsulated QDs (P-QDs) have shown a promising biocompatibility improvement, with minimal cytotoxicity in cells and animals, when compared to standard QDs [153, 154].
Furthermore, due to reviewers’ suggestions additional references were added and so the manuscripts references were renumber in rubric order.
Reviewer 2 Report
The manuscript by Montaseri et al. describes the use of porphyrin-based inorganic nanoparticles for cancer treatment. More specifically, the review is focused on PDT and PTT. Although the topic is interesting, the review in its current form is not appealing enough. The first sections are somehow messy and incomplete. Furthermore, authors should clearly explain which are the differences of this review with others already reported, dealing with the use of porphyrin-based nanomedicines for cancer treatment, i.e. Bioconjugate Chem. 2019, 30, 1585, Nanoscale, 2016, 8, 12394. Some comments and suggestions that may improve the quality of the manuscript are the following:
- The first three sections of the review are quite repetitive and somehow incomplete. For instance the mechanism of PDT and PTT are explained in the introduction, but could be expanded in these sections. In fact PTT is barely explained, so it is difficult to have a clear picture of their advantages when compared for instance with PTT performed with AuNPs alone.
- It is known that one of the main limitations of these therapies is the penetration of light into deep tissues. Authors could include a brief description of which lasers are commonly used, the potency, wavelength and penetration for instance.
- In section 3 authors talk about metalloporphyrins, metal-free, cationic and anionic porphyrins but they do not include any example. Something similar happens with the next classification, that is, first, second and third-generation porphyrins. In section 3 no examples can be found, while in section 4, some of them are included. I suggest merging both sections and including an extended description of phorphyrins, or at least to refer the reader to recent reviews.
- In section 4 it is stated, “Since NPs are small in size when conjugated to PSs they enhance passive diffusion into tumour 
cells”. This is really a vague description of NPs. As the review focuses on NPs, a better description of NPs with their intrinsic advantages should be included. In fact, one of the principal advantages is that they have high surface-to-volume ratios, which allows the functionalization of a higher number of biomolecules. This statement is included in magnetic nanoparticles, but not in other types. Also, authors should state why they have chosen inorganic NPs and not organic ones, or at least the advantages of using them. In the case of silica NPs it is difficult to understand the sentence: “which enables the formation of their desired surface structure within therapeutic 
applications”.
- Throughout the text, there are statements that are not currently supported by the literature. For instance:
- “nanoparticles (NPs) conjugated drugs, which have the ability to allow the 
drug to by-pass the immune system”: this is not accurate, as to date it is known that many NPs can be easily recognized by macrophages when injected.
- “improves their specificity to cancer cells and so eliminates off-target accumulation of drugs”: off-target effects are commonly decreased, but not totally eliminated. 

“Although chemotherapy, surgery and radiotherapy 
(when utilized singularly or in combination) are current conventional cancer treatments, they are problematic, since these treatment modalities exhibit nonspecific delivery, experience difficulties 
passing biological barriers and have low bio-distribution”: these drawbacks are true for chemotherapy, but not for surgery for instance, where no drug delivery is needed.
Author Response
Reviewer #2:
The manuscript by Montaseri et al. describes the use of porphyrin-based inorganic nanoparticles for cancer treatment. More specifically, the review is focused on PDT and PTT. Although the topic is interesting, the review in its current form is not appealing enough.
The first Sections are somehow messy and incomplete.
- The first Sections have been modified and expanded with some more statements to make them complete.
Furthermore, authors should clearly explain which are the differences of this review with others already reported, dealing with the use of porphyrin-based nanomedicines for cancer treatment, i.e. Bioconjugate Chem. 2019, 30, 1585, Nanoscale, 2016, 8, 12394.
Regarding the first reference, it has only investigated porphyrin-based organic NPs and the lack of inorganic based NPs motivated us to write this review. We have reviewed the second reference supplied and it does refer to some porphyrin-based inorganic NPs, however the referrals date back to studies performed in 2016 and this is out of the scope intent of this review since we structured it to only cover recent progress of porphyrin-based inorganic NPs over the last three years (2017 to 2020). This scope has been clearly stated in the abstract and conclusion of this article.
Lastly, the following paragraph with additional references and referral to reviewer suggested reference has been included in the conclusion of the manuscript:
- Various inorganic NPs have been scrutinized for porphyrin delivery. Former porphyrin loaded inorganic NPs in cancer treatment were evaluated by Zhou et al. (2016) [155] and studies by Xue et al. (2019) thoroughly reviewed porphyrin based organic NPs for cancer therapy [18]. Therefore, this review serves as a current update from these previous reviews, however with a core focus on inorganic porphyrin based nanomedicines within PDD, PTT and PDT application for cancer research.
Some comments and suggestions that may improve the quality of the manuscript are the following:
The first three Sections of the review are quite repetitive and somehow incomplete. For instance the mechanism of PDT and PTT are explained in the introduction, but could be expanded in these Sections. In fact PTT is barely explained, so it is difficult to have a clear picture of their advantages when compared for instance with PTT performed with AuNPs alone.
The mechanism of PDT has been moved to Sections 2, to eliminate repetitiveness and expanded for completeness.
- The mechanism of PDT involves the excitation of a photosensitizer (PS) at a specific wavelength [22]. The excited singlet state PS is then promoted to triplet state via intersystem crossing, which consequently transfers its energy to surrounding tissue molecular oxygen and so generates reactive oxygen species (ROS), which obliterates tumour cells via necrotic or apoptotic cell death pathways [22].
Additionally, the mechanism of PTT has been thoroughly explained as well in Section 2, for completeness by addition of the following paragraph and references:
- PTT is another cancer therapeutic treatment whereby a therapeutic agent is stimulated by both a specific wavelength of light and vibrational energy to produce heat through optical absorption, which consequently kills surrounding cancer cells [23, 24]. In PTT, near infrared laser (NIR) is used to irradiate tumour cells topically or interstitially to uniformly increase the temperature in the tumour, as well as prevent damage to healthy cells [23]. When the temperature in the centre of tumour reaches a therapeutic temperature above 50°C, effective ablation occurs [25, 26]. However, in some PTT cases this heat may leach out of the target tumour and so damage healthy surrounding cells. In order to circumvent this issue, photothermal absorbers are employed to elevate and concentrate the heat production in target tumour cells only [23]. Furthermore, photothermal absorbers require lowered light energy to reach therapeutic temperature and so only a small amount of heat escapes from the target tissue, preventing any damage to the heathy cells [23]. Various optical dyes such as photosensitizers (PSs) are can serve as potential photothermal absorbers, and so can be used to enhance the efficiency of phototherapy.
It is known that one of the main limitations of these therapies is the penetration of light into deep tissues. Authors could include a brief description of which lasers are commonly used, the potency, wavelength and penetration for instance.
The following paragraph additions and references have been added to the manuscript’s introduction:
- However, the depth of treatment relies on the wavelength of light that can activate a PS to generate reactive oxygen species (ROS). Shorter wavelengths (<650 nm) have a low penetration depth in tissues, while longer wavelengths above 850 nm are not sufficient for PS activation to generate excited singlet oxygen [13]. Therefore, the most appropriate phototherapeutic window index wavelength of PDT is within 650-850 nm range [13]. In addition, near infrared (NIR) light is the most suitable region in PTT treatment. In particular, NIR-II window (1000-1350 nm), compared to NIR-I window (750-950 nm) [14], since it can provide deep-seated tumour tissue penetration with high maximum permissible exposure to lasers (i.e. 1 W/cm2 at 1064 nm laser, 0.33 W/cm2 at 808 nm laser) [15].
Furthermore, the following paragraph and references have been included in Section 2:
- However, since the PDT and PTT therapeutic effect of porphyrins hinge on light energy activation, the application of porphyrin nanomedicines are limited to accessible tumours, such as skin or bladder cancer [29, 30]. Thus, researchers tend to incline towards the redshift porphyrin absorption beyond 700 nm, in order to enhance tissue penetration and overall phototherapeutic outcomes [18].
In Section, 3 authors talk about metalloporphyrins, metal-free, cationic and anionic porphyrins but they do not include any example. Something similar happens with the next classification, that is, first, second and third-generation porphyrins.
Examples of metalloporphyrins, metal-free, cationic and anionic porphyrins, as well as first, second and third-generation porphyrins have been included in Section 3 and the paragraph has been modified to include the following sentences with related references:
- Metalloporphyrins (porphyrins with metal ions within their inner cavity, such as Gd and Mn [35] are high contrast agents for MRI due to their high absorption coefficient [36]. Zn- and Al- porphyrins have been also employed in the field of PDT [35]. Cationic porphyrins (e.g. zinc(II)meso-tetrakis(N-n-hexylpyridinium-3-yl)porphyrin and 5-(4-trifluoromethylphenyl)-10,15,20-tris(4- trimethylammoniumphenyl) porphyrin) [34] are the most promising PSs for cancer treatment due to their amphiphilic properties [37], since anionic porphyrins such as tetrakis (4-sulfonatophenyl)-porphyrin and their derivatives tend to self-aggregate and so their overall ROS generation is self-limiting [38].
In Section 3 no examples can be found, while in Section 4, some of them are included. I suggest merging both Sections and including an extended description of porphyrins, or at least to refer the reader to recent reviews.
Examples of first, second and third generation porphyrin photosensitizers, their applications and limitations and related references have been all added in Section 3:
- PDT porphyrin based PSs (PBPs) are further classified into three different generations [31]. First-generation porphyrins (e.g. hematoporphyrin derivative (HpD) and photofrin) showed effective treatment in early clinical trials of PDT on brain [39], colorectal [40] and skin cancer [41]. However, the utilization of the first-generation porphyrins are limited due to their low extinction coefficient, water solubility, chemical impurity, photosensitive toxicity and suboptimal tissue penetration [17, 31]. Therefore, the second-generation porphyrins such as PBPs (e.g. lutrin, purlytin and tookad) were developed to address the drawbacks of first-generation ones. Despite many significant advantages of second-generation PBPs on melanoma [42] human prostate cancer [43] and breast cancer [44], high dark toxicity, limited cellular uptake and light penetration, as well as severe pain to patients were found [17, 31] and so this led the research towards investigating the third-generation porphyrins. Third-generation porphyrins, consist of certain second-generation porphyrin PSs conjugated to various targeting moieties such as antibodies, are the current focus of research. Third-generation PSs are basically conjugated to NPs and / or tumour targeting moieties, in order to overcome the limitations of first-and second-generation porphyrins, in relation to subcellular localization and targeted absorption and so overall allow for phototherapeutic efficiency enhancement [45-47].
- In addition, references #17, 31, 34 and 35 have been included in Section 4 to refer the readers to the recent studies.
In Section 4 it is stated, “Since NPs are small in size when conjugated to PSs they enhance passive diffusion into tumour cells”. This is really a vague description of NPs. As the review focuses on NPs, a better description of NPs with their intrinsic advantages should be included. In fact, one of the principal advantages is that they have high surface-to-volume ratios, which allows the functionalization of a higher number of biomolecules. This statement is included in magnetic nanoparticles, but not in other types.
Thank you for pointing this out. The following paragraph additions and references have been added to the Section 4 to further elaborate on the properties of NPs:
- The advent of NPs has allowed for the majority of conventional PS limitations to be overcome. NPs are small in size and range between 1 to 100 nm in diameter and some can contain /carry multiple theranostic agents [48]. Furthermore, their large surface to volume ratios can enhance the payload of PS delivered to target cells [49]. NPs can also prevent premature release of PS, which consequently can prevent nonspecific accumulation in healthy cells [50]. Importantly, NPs give amphilicity to PSs in order to bypass through the blood stream unaffected [51]. Since NPs are small in size when conjugated to PSs, they enhance passive diffusion into tumour cells through the EPR effect [4, 52].
In addition, the following paragraph has been added to the introduction:
- Therefore, many researches have shifted their focus into porphyrin-based nanomedicines [18, 19], since nanomedicines can enhance the bioavailability of therapeutic agents and so assist them to accumulate passively in tumours via the enhanced permeability and retention (EPR) effect [20]. Lastly, the multifunctionality of nanomedicines is of great value in order to assist in alleviating unwanted cancer treatment side effects [1].
Also, authors should state why they have chosen inorganic NPs and not organic ones, or at least the advantages of using them.
The following paragraph and references have been added to the Section 4:
- Drug delivery systems based on inorganic NPs provide the advantages of a wide surface area conjugation chemistry with versatile surface functionalization [53]. In addition, they bestow great biocompatibility, easy preparation and excellent physico-chemical properties in order to enhance therapeutic efficacy [53]. It is important to appreciate that non-biodegradable nanoparticles, including inorganic NPs, have recently drawn considerable research interest in the field of PDT, as multifunctional theranostic carriers, due to their unique optical properties and size, shape and porosity tunability, however is out of the scope of this review. Furthermore, when organic NPs are compared with inorganic polymeric biodegradable NPs, they do not degrade easily into biological systems and do not release a PS efficiently after activation, instead they allow oxygen species to seep out of the carrier. Thus, inorganic NPs (which is the focus of this review) are considered far more superior as they can carry many PSs to the tumour cells as a single nanoplatform, with far more beneficial outcomes [48].
In addition, the following paragraph has been added to the conclusion:
- Various inorganic NPs have been scrutinized for porphyrin delivery. Former porphyrin loaded inorganic NPs in cancer treatment were evaluated by Zhou et al. (2016) [155] and studies by Xue et al. (2019) thoroughly reviewed porphyrin based organic NPs for cancer therapy [18]. Therefore, this review serves as a current update from these previous reviews, however with a core focus on inorganic porphyrin based nanomedicines within PDD, PTT and PDT application for cancer research.
In the case of silica NPs it is difficult to understand the sentence: “which enables the formation of their desired surface structure within therapeutic applications”.
This sentence has been modified for clarity in Section 4.4 and an additional referenced sentence added:
- which facilitates the use of various methods of surface modifications for diagnostic and therapeutic applications [123, 124]. They also enable the fabrication of different silica nanoplatforms with various morphologies, sizes and porosity such as hollow or mesoporous silica NPs [125].
Throughout the text, there are statements that are not currently supported by the literature. For instance: the statements have been supported with related references.
“nanoparticles (NPs) conjugated drugs, which have the ability to allow the drug to by-pass the immune system”: this is not accurate, as to date it is known that many NPs can be easily recognized by macrophages when injected.
The following paragraph additions and references have been added to the introduction to clarify this statement:
- It has been demonstrated that the relationship between NPs and innate immune system responses (such as antigen-presenting cells or macrophage) depends on various parameters including size, shape, surface modification and charge, as well as hydrophobicity of NPs [5, 6]. For example, PEGylation of NPs can prevent non-specific uptake by innate immune system [7]. More importantly, therapeutic NPs require having immunosuppressive or anti-inflammatory properties to phagocytes in order to decrease the therapeutic dose and immune based side effects [5, 8]. For this reason, NPs conjugated with targeting ligands, such as antibodies show prolonged immunosuppressive effects [9] and improved specificity to cancer cells which consequently declines off-target accumulation of drugs [1].
“improves their specificity to cancer cells and so eliminates off-target accumulation of drugs”: off-target effects are commonly decreased, but not totally eliminated.
Noted. This sentence has been modified to state the following in the introduction:
- For this reason, NPs conjugated with targeting ligands, such as antibodies show prolonged immunosuppressive effects [9] and improved specificity to cancer cells which consequently declines off-target accumulation of drugs [1].
“Although chemotherapy, surgery and radiotherapy (when utilized singularly or in combination) are current conventional cancer treatments, they are problematic, since these treatment modalities exhibit nonspecific delivery, experience difficulties passing biological barriers and have low bio-distribution”: these drawbacks are true for chemotherapy, but not for surgery for instance, where no drug delivery is needed.
This paragraph has been modified with additional references in Section 2:
- Although chemotherapy and radiotherapy (when utilized singularly or in combination) are current conventional cancer treatments, they are problematic, since these treatment modalities exhibit nonspecific delivery, experience difficulties passing biological barriers and have low bio-distribution, thus their overall effectiveness is limited and they present additional unwanted side effects [1]. Furthermore, surgery is not considered a complete treatment for some kind of cancers and it is associated with anaesthesia, high blood loss and soft-tissue trauma [21].
Furthermore, due to reviewers’ suggestions additional references were added and so the manuscripts references were renumber in rubric order.
Reviewer 3 Report
The article "Recent Advances in Porphyrin based Inorganic Nanoparticles for Cancer Treatment" is a well written review outlining advances, current challenges and future perspectives in the niche of porphyrin based nanoparticles for cancer treatment. The article was well presented explaining the background and different studies carried out by scientific community using different types of Porphyrin nanoparticles for Cancer treatment.
Shortcomings/drawbacks:
- A few grammatical errors need to be fixed in the paper (Line 263, ardous/challenging should be used for difficulty; Lines 433- 437, sentence formation).
- Addition of more invivostudies using porphyrin based nanoparticles if any would have further increased the degree and significance of the article.
- Lines 82- 87: These lines explains three different classes of porphyrins for phototherapeutic applications. More background on the classification of porphyrins, their use and limitations for phototherapeutic applications is needed.
- Lines 204- 208: There is a transition to Two-photon PDT in this paragraph from PDT “off/on” state process in the previous paragraph which shows lack of continuity. This is evident in few more paragraphs (In line 51, PTAs were introduced suddenly from SDT earlier) in this article which needs to be rectified.
- Lines 374- 377: More background explaining the upconversion luminescence and anti-stokes process in regards to upconversion nanoparticles (UCNPs) is needed.
- Lines 430- 437: Despite mentioning that Cadmium QDs have higher toxicity and limited penetration of visible light for superficial tumours, author expresses contradicting statements like “application of cadmium free QDs as a promising area of research”. More clarity on what the author is trying to explain in this paragraph is needed.
Merits and Significance:
- Articles like these are essential for providing a summary of the niche like porphyrin based inorganic nanoparticles as a basis for further research to the community.
- The article provides an update to the research community about recent investigations of various types of Porphyrin based Nanoparticles approaches like Upconversion nanoparticles, Carbon nanoparticles, Noble Metallic nanoparticles, Magnetic nanoparticles, Quantum dots, and Silica nanoparticles.
Author Response
Reviewer #3:
The article "Recent Advances in Porphyrin based Inorganic Nanoparticles for Cancer Treatment" is a well written review outlining advances, current challenges and future perspectives in the niche of porphyrin based nanoparticles for cancer treatment. The article was well presented explaining the background and different studies carried out by scientific community using different types of Porphyrin nanoparticles for Cancer treatment.
Shortcomings/drawbacks:
A few grammatical errors need to be fixed in the paper (Line 263, ardous/challenging should be used for difficulty; Lines 433- 437, sentence formation).
In Section 4.2 the word “Difficulty” has been changed to “challenging”.
Lines 433-437 has been rephrased to the following paragraph in Section 4.6 with reference additions:
- The application of cadmium free QDs such as zinc and indium based QDs [148] or the conjugation of QDs to the surface or incorporation into polymeric NPs [149] are a promising area of research, since this enhances their overall application in clinical trials in order to investigate their overall effectiveness for cancer diagnosis and treatment [150]. However, due to the limited penetration of visible light, the utilization of both Cd containing and Cd-free QDs is restrained to superficial tumours for diagnosis of cancer only [1, 151]. Nevertheless, QDs possess a high two-photon absorption (TPA) cross-section (σ~104 GM) [152], which can be employed in two-photon bioimaging [153]. Furthermore, polymer-encapsulated QDs (P-QDs) have shown a promising biocompatibility improvement, with minimal cytotoxicity in cells and animals, when compared to standard QDs [153, 154].
Addition of more invivo studies using porphyrin based nanoparticles if any would have further increased the degree and significance of the article.
The application of porphyrin based inorganic NPs for in vivo treatment of cancer have been already mentioned in the references #69, 81, 90, 116, 135, 137, 145 and 147 within the manuscript, we have highlighted them and included more information for clarity that they are in fact in vivo studies.
Lines 82- 87: These lines explains three different classes of porphyrins for phototherapeutic applications. More background on the classification of porphyrins, their use and limitations for phototherapeutic applications is needed.
These lines have been modified to and moved to the following paragraphs in Section 3 with additional references and explanation for further clarity in relation to the application and limitations of each generation:
- PDT porphyrin based PSs (PBPs) are further classified into three different generations [31]. First-generation porphyrins (e.g. hematoporphyrin derivative (HpD) and photofrin) showed effective treatment in early clinical trials of PDT on brain [39], colorectal [40] and skin cancer [41]. However, the utilization of the first-generation porphyrins are limited due to their low extinction coefficient, water solubility, chemical impurity, photosensitive toxicity and suboptimal tissue penetration [17, 31]. Therefore, the second-generation porphyrins such as PBPs (e.g. lutrin, purlytin and tookad) were developed to address the drawbacks of first-generation ones. Despite many significant advantages of second-generation PBPs on melanoma [42] human prostate cancer [43] and breast cancer [44], high dark toxicity, limited cellular uptake and light penetration, as well as severe pain to patients were found [17, 31] and so this led the research towards investigating the third-generation porphyrins. Third-generation porphyrins, consist of certain second-generation porphyrin PSs conjugated to various targeting moieties such as antibodies, are the current focus of research. Third-generation PSs are basically conjugated to NPs and / or tumour targeting moieties, in order to overcome the limitations of first-and second-generation porphyrins, in relation to subcellular localization and targeted absorption and so overall allow for phototherapeutic efficiency enhancement [45-47].
Lines 204- 208: There is a transition to Two-photon PDT in this paragraph from PDT “off/on” state process in the previous paragraph which shows lack of continuity.
The following paragraph additions and references have been added in Section 4.1 before 2P-PDT to make it continuous:
- Generally, phototherapeutic agents are mainly locate in adjacent regions of a tumour, due to their abnormal vasculature nature and high interstitial fluid pressure (IFP), thus large parts of tumour remain unaffected, since they don’t adequately absorb PSs [82]. However, evidence has shown that many types of important cells associated with tumour initiation and progression are those fed by the defective blood vessels which supply solid tumors [83, 84] and research must focus on phototherapy research to target these cells, which in turn can prevent tumor growth and metastases [85]. Various strategies have focused on adjacent cells, while those distant from blood vessels remain untouched [86]. Therefore, the development of novel strategies to improve the penetration and uptake properties of PSs still remains an obstacle in nanotherapeutics.
This is evident in few more paragraphs (In line 51, PTAs were introduced suddenly from SDT earlier) in this article which needs to be rectified.
This paragraph has been modified to the following for continuity in Section 4.1:
- A new treatment modality known as sonodynamic therapy (SDT) was proposed by Liang et.al. (2019). SDT is used to excite and so activate PSs to produce high enough ROS levels for effective cancer therapy [90]. Furthermore, as already mentioned, photothermal absorbers or photothermal transduction agents (PTAs) are adopted in PTT to harvest light energy and generate hyperthermia [91]. These PTAs are categorized into semiconductor nanocrystals (NCs) [92], inorganic materials [93] and organic dyes [94]. It has been reported that platinum or palladium-based photothermal transduction agents (PTAs) as inorganic PTAs have a higher photothermal stability and better catalytic properties than when compared to Au based PTAs [95, 96]. Furthermore, studies have noted that copper sulfide semiconductor NCs provide very high LSPR within the NIR region [97]. In this regard, studies developed platinum-copper sulfide Janus nanoparticles conjugated to tetra-(4-aminophenyl) porphyrin (TAPP) to overcome the low penetration depth of PDT [90]. In fact, they integrated semiconductor NCs and noble platinum metal to form their PTAs [90].
Lines 374- 377: More background explaining the upconversion luminescence and anti-stokes process in regards to upconversion nanoparticles (UCNPs) is needed.
The following paragraph additions and references have been added to the Section 4.5, for more background explanation:
- Generally, the conversion of NIR light to a shorter wavelength of light is known as upconversion, which is an anti-Stokes process [140, 141]. Upconversion luminescence is a nonlinear process whereby successive lower energy photons absorb luminescence and so emit higher energy photons [140, 142]. This process has the advantage of low light scattering and autofluorescence background, as well as high tissue penetration depth, since this excitation occurs in the NIR region which is located within phototherapeutic window. This also consequently decreases any photo damage biological tissues might experience [143].
Lines 430- 437: Despite mentioning that Cadmium QDs have higher toxicity and limited penetration of visible light for superficial tumours, author expresses contradicting statements like “application of cadmium free QDs as a promising area of research”. More clarity on what the author is trying to explain in this paragraph is needed.
This paragraph has been modified in Section 4.6, for further clarity:
- Despite the great properties QDs possess in terms of high tunability with high quantum yields [1], there are limited studies regarding their application in porphyrin based PDT cancer treatments. The reason behind the limitation of QD porphyrin based PDT applications, is due to the high toxicity they possess, since most consist of toxic heavy metals like cadmium ions [1]. The application of cadmium free QDs such as zinc and indium based QDs [148] or the conjugation of QDs to the surface or incorporation into polymeric NPs [149] are a promising area of research, since this enhances their overall application in clinical trials in order to investigate their overall effectiveness for cancer diagnosis and treatment [150]. However, due to the limited penetration of visible light, the utilization of both Cd containing and Cd-free QDs is restrained to superficial tumours for diagnosis of cancer only [1, 151]. Nevertheless, QDs possess a high two-photon absorption (TPA) cross-section (σ~104 GM) [152], which can be employed in two-photon bioimaging [153]. Furthermore, polymer-encapsulated QDs (P-QDs) have shown a promising biocompatibility improvement, with minimal cytotoxicity in cells and animals, when compared to standard QDs [153, 154].
Furthermore, due to reviewers’ suggestions additional references were added and so the manuscripts references were renumber in rubric order.
Round 2
Reviewer 1 Report
The changes are helpful.
Please clarify re. the Soret band-
"Their ability to auto-fluoresce within the 400 to 440 nm 120 wavelength region (Soret band) [33]"
excitation would be 400-440. Where is emission?
The language usage is still poor. If anything, the copy edit made it worse. A technical copy editor should be employed who knows how to properly pluralize scientific terms. There are many, many occasions of poor pluralization and subject-verb disagreement.
Just the first paragraph:
Despite some major breakthroughs in cancer treatment, cancer is still one of the most significant 23 causes of death in patients, after heart and infectious diseases [1].
>>Redundant wording. Better as "Despite some major breakthroughs in treatment, cancer is still one of the most significant causes of death , after heart and infectious diseases [1].
Current disadvantages of 24 conventional cancer therapies, such as Chemotherapy and Radiation,
>> no capital letters should be used on Chemotherapy and Radiaiton.
are nonspecific drug delivery 25 mode of actions, poor bio-distribution and biological barriers clearance, which limit their overall 26 effectiveness [2, 3].
>>poor plurals. "Modes of action", "biological barrier clearance"
These recurring errors make the manuscript hard to read.
Reviewer 2 Report
The quality of the review has improved substantially after the changes made by the authors. However, I feel that there are still some sentences that are not clear or accurate and authors should be more careful when using them:
- Page 1, line 39: “NPs require having immunosuppressive or anti-inflammatory properties to 
phagocytes” : All NPs have to be immunosuppressive or anti-inflammatory or only in some occasions ? It is known that the vast majority of the NPs do not have these properties.
- Line 168: “Furthermore, when organic NPs are 
compared with inorganic polymeric biodegradable NPs, they do not degrade easily into biological 
systems and do not release a PS efficiently after activation” : what do the authors mean by inorganic polymeric biodegradable NPs? Not all types of inorganic Nps are biodegradable. On the other side, many organic NPs are easily biodegradable. Similarly, the sentence “inorganic NPs (which is the focus of this review) are considered far more superior as they can carry many PSs to the tumour cells as a single nanoplatform, with far more 
beneficial outcomes” have to be clarified. Organic NPs can also carry many PSs to the tumour cells. 

- Line 343: All AuNPs have a two-distinctive absorption bands?
Minor changes:
-abbreviations should be revised. For instance ROS and NIR now appear before putting the abbreviation.
-some sentences have errors: Various optical dyes such as photosensitizers (PSs) are can serve as 
potential photothermal absorbers, and so can be used to enhance the efficiency of phototherapy.